# Cardiometabolic Index, BMI, Waist Circumference, and Cardiometabolic Multimorbidity Risk in Older Adults

**DOI:** 10.3390/geriatrics11010004

**Published:** 2025-12-30

**Authors:** Setor K. Kunutsor, Jari A. Laukkanen

**Affiliations:** 1Section of Cardiology, Department of Internal Medicine, Max Rady College of Medicine, Rady Faculty of Health Sciences, University of Manitoba, Winnipeg, MB R2H 2A6, Canada; 2Leicester Real World Evidence Unit, Diabetes Research Centre, University of Leicester, Leicester General Hospital, Gwendolen Road, Leicester LE5 4WP, UK; 3Institute of Clinical Medicine, Department of Medicine, University of Eastern Finland, FI-70210 Kuopio, Finland; jariantero.laukkanen@uef.fi; 4Wellbeing Services County of Central Finland, Department of Medicine, 40620 Jyväskylä, Finland

**Keywords:** cardiometabolic index, body mass index, waist circumference, cardiometabolic multimorbidity, cohort study

## Abstract

**Background/Objectives:** The cardiometabolic index (CMI) is a simple anthropometric–metabolic indicator that has recently gained attention as a marker of cardiometabolic risk. This study compared the associations and predictive utility of CMI, body mass index (BMI), and waist circumference (WC) for cardiometabolic multimorbidity (CMM). **Methods:** Data were drawn from 3348 adults (mean age 63.5 years; 45.1% male) in the English Longitudinal Study of Ageing who were free of hypertension, coronary heart disease, diabetes, and stroke at wave 4 (2008–2009). CMI was calculated using the triglyceride-to-HDL-cholesterol ratio and the waist-to-height ratio. Incident CMM at wave 10 (2021–2023) was defined as the presence of ≥2 of these conditions: hypertension, cardiovascular disease, diabetes, or stroke. Odds ratios (ORs) with 95% confidence intervals (CIs) and measures of discrimination were estimated. **Results:** During 12–15 years of follow-up, 197 CMM cases were recorded. CMI, BMI, and WC were each linearly related to CMM. Higher CMI was associated with increased CMM risk (per 1-SD increase: OR 1.25, 95% CI 1.08–1.44; highest vs. lowest tertile: OR 1.88, 95% CI 1.09–3.25), with similar effect sizes for BMI. WC showed stronger associations (per 1-SD increase: OR 1.46, 95% CI 1.25–1.71; highest vs. lowest tertile: OR 2.16, 95% CI 1.35–3.44). Adding CMI to a base model resulted in a small, non-significant improvement in discrimination (ΔC-index = 0.0032; *p* = 0.55) but significantly improved model fit (−2 log-likelihood *p* = 0.004), with comparable effects for BMI and greater improvements for WC. **Conclusions:** In this older UK cohort, higher CMI levels were associated with increased long-term risk of CMM but did not outperform traditional adiposity measures such as BMI and WC.

## 1. Introduction

Cardiometabolic diseases, including cardiovascular disease (CVD), type 2 diabetes, hypertension, and metabolic syndrome, remain leading causes of morbidity, disability, and premature mortality worldwide [1]. Their prevalence continues to rise, driven by population ageing, lifestyle changes, and increasing adiposity. Numerous risk factors for cardiometabolic diseases are well established, including unhealthy diet, physical inactivity, tobacco use, dyslipidemia, and elevated blood pressure [2]. However, one of the most important modifiable determinants is excess adiposity, with overweight and obesity representing its most common clinical manifestations. These conditions are strongly associated with a wide range of adverse cardiometabolic outcomes and play a central role in disease progression and mortality [3]. More recently, the concept of cardiometabolic multimorbidity (CMM), defined as the co-occurrence of two or more major cardiometabolic conditions, has gained prominence as a critical public health challenge [4]. Individuals with CMM experience accelerated disease progression, greater healthcare utilization, reduced quality of life, and markedly elevated risk of disability and premature mortality [5]. As the global burden of cardiometabolic disorders grows, so too does the need for tools that enhance early identification of individuals at high risk of developing multimorbidity.

Obesity, particularly abdominal or visceral obesity, plays a central etiological role in the development of cardiometabolic diseases [3]. Although body mass index (BMI) is widely used to assess general adiposity and is consistently associated with adverse outcomes, it is an imperfect measure. Body mass index does not distinguish between fat and lean mass, varies according to age, sex, and ethnicity, and does not reflect body fat distribution [6,7]. Individuals with normal-weight obesity—characterized by normal BMI but elevated visceral adiposity—may be misclassified as low risk despite having high cardiometabolic vulnerability. Measures of central adiposity such as waist circumference (WC) better capture fat distribution and often outperform BMI in predicting cardiometabolic risk [8]. Nonetheless, WC is strongly correlated with BMI and body size and may be confounded by height [9]. To address these limitations, several newer anthropometric indices have been developed to more accurately characterize abdominal adiposity and visceral fat deposition. These include A Body Shape Index (ABSI) [10], Body Roundness Index (BRI) [11], Visceral Adiposity Index (VAI) [12], Weight-Adjusted Waist Index (WWI) [13], Lipid Accumulation Product (LAP) [14], Relative Fat Mass (RFM) [15], and the Conicity Index (CONI) [16]. A growing body of evidence indicates that many of these novel indices have superior predictive performance for adverse cardiometabolic outcomes compared with BMI and WC [17,18,19]. In addition, associations between several of these indices and CMM have been reported [17,20,21]. The Cardiometabolic Index (CMI), a simple anthropometric–metabolic indicator combining the triglyceride-to-high density lipoprotein cholesterol (HDL-C) ratio with the waist-to-height ratio [22], has recently emerged as a practical marker of visceral adiposity, insulin resistance, and cardiometabolic risk [23,24,25]. By integrating a measure of central adiposity with lipid metabolism, CMI is designed to capture both fat distribution and metabolic dysfunction—key pathways linking visceral fat to cardiometabolic disease, particularly in older adults in whom age-related changes in lipid profiles and widespread use of lipid-lowering therapies may limit the discriminatory value of anthropometric measures alone. Early studies suggest that CMI demonstrates stronger associations with and enhanced discriminative ability for cardiometabolic diseases relative to traditional adiposity measures [26,27].

Despite this emerging interest, evidence on the relationship between CMI and CMM remains limited. To date, no study has assessed the prospective association between CMI and incident CMM in a general population, nor has any investigated the predictive utility of CMI for identifying individuals at elevated risk of developing multimorbidity. Addressing this evidence gap is important, given the growing burden of CMM and the need for simple, scalable markers that can improve risk assessment in clinical and population health settings. The English Longitudinal Study of Ageing (ELSA), a large, nationally representative, prospective cohort of middle-aged and older adults in England, offers a unique opportunity to evaluate these questions. Therefore, the aim of this study is to assess the nature and magnitude of the association between CMI and the risk of CMM, evaluate the ability of CMI to predict incident CMM, and compare its performance with BMI and WC within the same sample of participants. We hypothesized that higher levels of the CMI would be independently associated with an increased risk of incident CMM and would provide greater predictive utility for CMM compared with traditional anthropometric measures such as BMI and WC.

## 2. Materials and Methods

### 2.1. Study Population

This investigation followed the Strengthening the Reporting of Observational Studies in Epidemiology (STROBE) recommendations (Appendix A). The study utilized a prospective cohort design. Data were drawn from ELSA, an ongoing population-based cohort of male and female adults aged 50 years and older who were living in private residential households in England [28]. The cohort was originally assembled from participants in the 1998, 1999, and 2001 Health Survey for England rounds. The ELSA cohort began in 2002–2003 with 11,391 respondents and has since collected data every two years in successive waves [28]. For the purposes of this study, data from wave 4 (2008–2009) were treated as the baseline, and participants were followed until wave 10 (2021–2023). We excluded individuals who had a history of hypertension, coronary heart disease, diabetes, or stroke at baseline [17,21]. Participants also had to have complete information on CMI, BMI, WC, all covariates, and CMM outcomes. After applying these criteria, 3348 men and women were eligible for inclusion in the final analytical sample (Appendix A). All respondents provided written informed consent, and ethical approval for ELSA was obtained from the London Multicentre Research Ethics Committee (reference number: MREC/01/2/91).

### 2.2. Exposures, Covariates, and Outcome

Demographic, behavioural, and clinical information was collected using a combination of interviewer-led assessments and self-completed questionnaires, both of which have been extensively used in previous ELSA waves. These instruments captured a wide range of participant characteristics, including age, sex, lifestyle behaviours, and past medical history [20,28,29]. Alcohol use was determined from responses to a question asking how frequently individuals consumed alcoholic beverages over the previous year. Tobacco exposure was assessed in two stages: participants were first asked whether they had ever smoked, and those responding “yes” were subsequently questioned about their current smoking behaviour [29]. Usual physical activity levels were obtained through a validated questionnaire documenting engagement in light, moderate, and vigorous leisure-time activities. In line with ELSA conventions, respondents were classified into one of four categories—physically inactive (no moderate or vigorous activity), low (light activity only), moderate (moderate activity at least once weekly), or high (vigorous activity at least once weekly) [28,30,31]. Standardized clinical examinations were performed by trained nurses at Mobile Examination Centers. These examinations included anthropometric measurements (height, weight, and waist circumference), blood pressure assessment, venous blood collection, and tests of physical function, such as handgrip strength (HGS). Body weight was recorded to the nearest 0.1 kg using calibrated electronic scales with participants wearing light clothing and no shoes. Waist circumference was measured at the midpoint between the iliac crest and the lower costal margin to the nearest even millimetre [32]. Handgrip strength was evaluated using a Smedley dynamometer (Stoelting Co., Wood Dale, IL, USA). Participants completed six trials—three per hand—starting with the dominant hand and alternating sides, with approximately one minute of rest between attempts. The maximum value obtained across all trials was used for analysis [33]. Body mass index was computed by dividing weight in kilograms by height in metres squared (kg/m^2^). Cardiometabolic Index was calculated using the formula: (Triglycerides [mmol/L] ÷ HDL-C [mmol/L]) × (WC [cm] ÷ height [cm]) [22]. Blood pressure measurements were taken on the right arm using an automated Omron HEM 907 device (OMRON Healthcare Europe BV, Hoofddorp, The Netherlands). Three readings were recorded, and the average of the second and third was used for analyses [32]. Participants’ health conditions, including hypertension, coronary heart disease, diabetes, and stroke, were obtained from participants’ reports of prior physician diagnoses collected during face-to-face assessments using structured questionnaires [29]. In the context of ELSA, CVD was operationalized based on participant reports of a prior diagnosis of “other heart disease”, with “heart attack” and stroke considered as separate outcomes. Because outcomes were self-reported, more detailed clinical definitions or adjudication of CVD subtypes were not available. However, validation studies conducted within ELSA and its sister study - the Health and Retirement Study - have shown moderate concordance between self-reported diagnoses and clinically verified records for several cardiometabolic conditions [32,34]. Cardiometabolic multimorbidity at wave 10 was defined as the presence of at least two of the following: hypertension, CVD, diabetes, or stroke [17,21].

### 2.3. Statistical Analysis

We first summarized the baseline characteristics of participants using standard descriptive approaches. Continuous variables were presented either as means with standard deviations (SDs) or as medians with interquartile ranges (IQRs), depending on their distribution as assessed through graphical methods and normality tests. Categorical characteristics were reported as frequencies and percentages. Group differences were examined using *t*-tests or Mann–Whitney tests for continuous variables, and chi-square tests for categorical variables. To investigate the potential nonlinearity of the associations of CMI, BMI, and WC with CMM risk, we fitted multivariable logistic regression models incorporating restricted cubic splines (RCS). Knots were placed following Harrell’s recommendations for samples of this size, located at the 5th, 35th, 65th, and 95th percentiles of each exposure distribution [35]. Nonlinearity was evaluated using likelihood ratio tests comparing spline-enhanced models with models containing only a linear term [36,37]. Multivariable logistic regression was used to compute odds ratios (ORs) with 95% confidence intervals (CIs). Three stepwise models were specified: Model 1: age and sex; Model 2: Model 1 plus smoking status, alcohol intake, systolic blood pressure (SBP), total cholesterol, HDL-C, and HGS; and Model 3: all previous covariates with physical activity added. These covariates were selected because of their established links with cardiometabolic disease, their use in prior ELSA investigations [17,20,21,30,38], and their potential confounding role in the exposures of interest. The exposures (CMI, BMI, and WC) were modelled per 1-SD increment and as tertiles. To evaluate the predictive contribution of CMI, BMI, and WC, we estimated Harrell’s C-indices for models including conventional risk factors (age, sex, smoking, alcohol consumption, SBP, total cholesterol, HDL-C, HGS, and physical activity) and then compared these with models additionally incorporating each adiposity metric. Changes in C-indices were tested using DeLong’s method [39]. Because discrimination alone may not fully capture improvements in prediction, we also compared model fit using changes in the −2 log-likelihood statistic, an approach recommended for assessing the incremental value of new biomarkers [40]. All analyses were conducted using Stata MP version 18.0 (StataCorp, College Station, TX, USA).

## 3. Results

### 3.1. Baseline Characteristics

Table 1 presents the baseline characteristics of the study’s participants overall and by incident CMM status. The mean (SD) age of the 3348 study participants at baseline was 63.5 (8.5) years, with ages ranging from 50 to 99 years. Males constituted 45.1% of the cohort. The mean (SD) of CMI, BMI, and WC was 0.70 (0.59), 27.6 (4.9) kg/m^2^, and 95.0 (13.3) cm, respectively. Participants who developed CMM at end of follow-up period had higher levels of CMI and anthropometric indices (BMI, WC, height, and weight), higher levels of HGS, SBP, and triglycerides, and lower levels of total cholesterol and HDL-C.

### 3.2. Associations of CMI, BMI, and WC with CMM

During the 12–15 years of follow-up, 197 cases of CMM were recorded. Restricted cubic spline models adjusted for relevant covariates showed that the association between CMI and CMM followed an approximately linear pattern, with no evidence of nonlinearity (*p* = 0.25). The risk of CMM increased progressively across the spectrum of CMI values (Figure 1A). In multivariable models accounting for age, sex, lifestyle factors, SBP, lipids, and HGS, each 1-SD rise in CMI was associated with higher odds of CMM (OR 1.26, 95% CI 1.09–1.45). Additional adjustment for physical activity produced similar results (OR 1.25, 95% CI 1.08–1.44) (Table 2). When CMI was examined in tertiles, participants in the highest group had substantially greater risk compared with those in the lowest group, with adjusted ORs of 1.98 (95% CI 1.15–3.41) and 1.88 (95% CI 1.09–3.25) in Models 2 and 3, respectively.

Body mass index demonstrated a similarly linear dose–response association with CMM (*p* = 0.68), with risk increasing steadily across BMI values from 21 to 48 kg/m^2^ (Figure 1B). In fully adjusted models (Model 2), the odds of developing CMM rose by 31% per 1-SD increase in BMI (OR 1.31, 95% CI 1.14–1.49). Further adjustment for physical activity resulted in modest attenuation (OR 1.28, 95% CI 1.12–1.47) (Table 2). Comparisons of the highest versus lowest BMI tertiles yielded adjusted ORs of 1.95 (95% CI 1.29–2.95) and 1.88 (95% CI 1.24–2.85) in Models 2 and 3, respectively.

Waist circumference showed the strongest associations of the three adiposity indices. Spline analyses again indicated a linear relationship (*p* = 0.70), with CMM risk increasing sharply as WC rose from approximately 89.5 to 145.5 cm (Figure 1C). In Model 2, each 1-SD increment in WC was associated with a 49% higher odds of CMM (OR 1.49, 95% CI 1.28–1.73), with minimal attenuation after further adjustment for physical activity (OR 1.46, 95% CI 1.25–1.71). Tertile comparisons showed that participants in the highest WC category had more than double the risk of CMM compared with those in the lowest category, with adjusted ORs of 2.27 (95% CI 1.43–3.60) and 2.16 (95% CI 1.35–3.44) for Models 2 and 3, respectively (Table 2).

### 3.3. Risk Prediction

Table 3 presents the findings from the prediction analyses. The reference model containing conventional cardiometabolic risk factors yielded a C-index of 0.6892 (95% CI: 0.6500, 0.7285). When CMI was added to this model, the C-index increased slightly to 0.6924 (95% CI: 0.6528, 0.7319), representing a small and non-significant improvement of 0.0032 (*p* = 0.55). Despite the modest change in discrimination, model fit improved significantly, as shown by the −2 log-likelihood statistic (*p* = 0.004). Including BMI in the baseline model led to a similarly small increase in discrimination (ΔC-index = 0.0049, *p* = 0.46), while again producing a significant enhancement in model fit (*p* < 0.001). Adding WC produced the largest increment in the C-index (ΔC-index = 0.0100, *p* = 0.24), and the −2 log-likelihood test indicated a clear improvement in overall fit (*p* < 0.001).

## 4. Discussion

In this national representative cohort of older adults, several key observations emerged. First, individuals who developed CMM over follow-up had less favourable baseline profiles, including higher levels of CMI, BMI, WC, weight, blood pressure, and triglycerides, alongside lower concentrations of HDL-C. Second, CMI, BMI, and WC each demonstrated clear linear dose–response relationships with incident CMM, with the steepest gradient observed for WC. Third, higher values of all three adiposity measures were independently associated with greater CMM risk, although WC consistently showed the strongest effect sizes in the same analytic sample. Finally, adding CMI, BMI, or WC to a model containing established cardiometabolic risk factors produced small, non-significant improvements in discrimination, but each measure significantly enhanced overall model fit, with WC showing the greatest incremental contribution.

Several prior investigations have evaluated the CMI in relation to individual adverse outcomes, including CVD, hypertension, type 2 diabetes, and metabolic dysfunction [41,42,43]. Across these studies, higher CMI has consistently been linked to elevated cardiometabolic risk, supporting its role as an indicator of visceral adiposity and metabolic derangement. However, despite this growing body of evidence, no published research to date has prospectively examined the relationship between CMI and the development of CMM in a general population cohort, nor has any study assessed the predictive value of CMI specifically for CMM. The only related evidence we identified was a cross-sectional analysis of U.S. adults with non-alcoholic fatty liver disease, which reported a positive association between CMI and prevalent CMM, following a nonlinear pattern [44]. While consistent with our observed direction of effect, that study’s high-risk clinical sample, cross-sectional design, and limited generalizability restrict its applicability to broader populations. In this context, the present study fills an important evidence gap. By evaluating CMI in a large, community-based cohort using a prospective design and directly comparing its performance with BMI and WC, our findings provide the first population-level evidence on the role of CMI in predicting incident CMM.

Several factors may help explain why CMI did not outperform BMI or WC in this cohort, despite evidence from previous studies suggesting its superiority for predicting cardiometabolic risk [26,27]. First, the study population consisted of older adults, among whom age-related changes in body composition, such as sarcopenia, fat redistribution, and declines in muscle mass, may weaken the discriminatory value of lipid-derived indices like CMI. In later life, anthropometric markers that more directly capture central adiposity, such as WC, may better reflect cardiometabolic vulnerability than composite indices incorporating triglycerides and HDL-C [45,46], which themselves are influenced by ageing, comorbidities, and medication use. Second, the relatively low variation in lipid profiles in an ageing population with potentially high use of lipid lowering therapy [47] may have reduced the incremental contribution of the triglyceride-to-HDL ratio, a key component of CMI. Finally, differential measurement error across the indices—including single-time biochemical assessments versus repeated anthropometric measures—may also contribute to the stronger performance of WC.

The results of this study suggest that while emerging adiposity markers such as CMI may offer additional metabolic insight, traditional anthropometric measures (particularly WC) remain highly relevant for risk assessment in certain demographic groups, including older adults. The stronger associations observed for WC indicate that simple measures of central adiposity continue to provide robust clinical information, even in the context of newer composite indices. These findings also reinforce that no single adiposity measure is universally optimal across all populations. Age-related physiological changes, medication use, and varying patterns of fat distribution may influence the performance of different indices [48]. Accordingly, clinicians and researchers should consider demographic context, population characteristics, and measurement feasibility when selecting adiposity metrics for cardiometabolic risk evaluation.

This study has several notable strengths. It is, to our knowledge, the first to prospectively examine the relationship between CMI and incident CMM and to directly compare the predictive value of CMI with BMI and WC in the same general population. The analytical approach was robust, incorporating spline modelling to assess dose–response patterns, extensive adjustment for established cardiometabolic risk factors, and formal evaluation of predictive performance. Several limitations should also be considered. The observational design prevents causal inference, and reliance on self-reported diagnoses and lifestyle factors may introduce misclassification or recall bias. Despite extensive multivariable adjustment, residual confounding cannot be excluded, including confounding arising from measurement error in included variables and from relevant unmeasured factors. We were unable to account for several potentially important covariates, such as kidney function, uric acid levels, and biomarkers of insulin resistance. In addition, direct measures of cardiorespiratory fitness and muscle mass were not available, which are independent predictors of cardiometabolic outcomes [49,50]. However, we adjusted for self-reported physical activity as a proxy for cardiorespiratory fitness and for HGS, which may partially mitigate the impact of these unmeasured factors. Handgrip strength is widely recognized as a simple and robust indicator of overall muscle strength, physical function, and biological ageing in older adults [51]. In this cohort, the mean HGS was approximately 31 kg, a value that is generally above commonly used thresholds for functional dependence and sarcopenia in community-dwelling older populations. Handgrip strength thresholds commonly used to indicate probable sarcopenia are <27 kg for men and <16 kg for women, as recommended by the European Working Group on Sarcopenia in Older People (EWGSOP2) [52]. This suggests that, on average, participants had preserved muscular strength at baseline. Nevertheless, lower HGS has been consistently linked to increased cardiometabolic risk, frailty, and adverse health outcomes [51,53], and its inclusion in our models helped account for underlying differences in physical fitness and muscle health. The observed associations between adiposity measures and CMM were therefore independent of baseline muscular strength, underscoring the relevance of adiposity-related risk beyond sarcopenia-related vulnerability in later life. The absence of diagnosis dates limited the use of time-to-event methods. The cohort consisted of older English adults, which may restrict generalizability to younger individuals or other demographic groups. Finally, the relatively small number of CMM events may have reduced precision in some estimates and limited subgroup analyses. These findings should be interpreted with these limitations in mind, and further studies in more diverse and larger populations are warranted.

## 5. Conclusions

In this cohort of older UK adults, CMI, BMI, and WC were each linearly associated with higher risk of CMM and provided modest incremental value beyond traditional risk factors. Waist circumference showed the strongest associations and contributed most to model performance, indicating that traditional measures of central adiposity remain highly informative for risk assessment in older populations. Further work in more diverse populations is needed to confirm these findings.

## Figures and Tables

**Figure 1 geriatrics-11-00004-f001:**
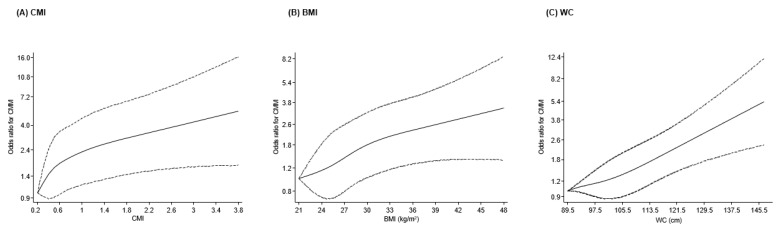
Spline curves of the associations of CMI, BMI, and WC with the risk of cardiometabolic multimorbidity. (**A**) CMI (**B**) BMI (**C**) WC. CMI, Cardiometabolic Index; BMI, body mass index; WC, waist circumference. Dashed lines represent the 95% confidence intervals for the spline model (solid line). The model was adjusted for age, sex, smoking status, alcohol consumption, systolic blood pressure, total cholesterol, high-density lipoprotein cholesterol, handgrip strength, and physical activity.

**Table 1 geriatrics-11-00004-t001:** Baseline characteristics overall and according to incident CMM status at end of follow-up period.

Characteristic	Overall (N = 3348)	No CMM (N = 3151)	Yes CMM (N = 197)	*p*-Value
	Mean (SD), Median (Q1–Q3), or n (%)	Mean (SD), Median (Q1–Q3), or n (%)	Mean (SD), Median (Q1–Q3), or n (%)	
Cardiometabolic index	0.70 (0.59)	0.69 (0.57)	0.95 (0.74)	<0.001
Body mass index, kg/m^2^	27.6 (4.9)	27.4 (4.8)	29.8 (5.6)	<0.001
Waist circumference, cm	95.0 (13.3)	94.6 (13.0)	101.8 (15.2)	<0.001
Height, cm	166.7 (9.5)	166.6 (9.5)	168.1 (9.6)	0.028
Weight, kg	76.7 (15.6)	76.2 (15.4)	84.3 (18.1)	<0.001
Age, yrs	63.5 (8.5)	63.5 (8.6)	62.6 (6.5)	0.13
Sex				0.10
Male	1510 (45.1%)	1410 (44.7%)	100 (50.8%)	
Female	1838 (54.9%)	1741 (55.3%)	97 (49.2%)	
Current smoker				0.034
No	2873 (85.8%)	2714 (86.1%)	159 (80.7%)	
Yes	475 (14.2%)	437 (13.9%)	38 (19.3%)	
Alcohol categories				0.85
None	1121 (33.5%)	1051 (33.4%)	70 (35.5%)	
1–2 times/wk	843 (25.2%)	797 (25.3%)	46 (23.4%)	
3–4 times/wk	636 (19.0%)	601 (19.1%)	35 (17.8%)	
5 or more times/wk	748 (22.3%)	702 (22.3%)	46 (23.4%)	
Handgrip strength, kg	31.0 (11.4)	30.9 (11.3)	32.7 (11.9)	0.028
SBP, mmHg	130 (17)	130 (17)	137 (17)	<0.001
Total cholesterol, mmol/L	5.84 (1.14)	5.85 (1.13)	5.59 (1.20)	0.002
HDL cholesterol, mmol/L	1.59 (0.42)	1.60 (0.42)	1.45 (0.41)	<0.001
Triglyceride, mmol/L	1.40 (1.00, 2.00)	1.40 (1.00, 2.00)	1.70 (1.20, 2.50)	<0.001
Physical activity level				0.18
Physically inactive	91 (2.7%)	85 (2.7%)	6 (3.0%)	
Low	593 (17.7%)	547 (17.4%)	46 (23.4%)	
Moderate	1810 (54.1%)	1710 (54.3%)	100 (50.8%)	
High	854 (25.5%)	809 (25.7%)	45 (22.8%)	

CMM, cardiometabolic multimorbidity; HDL, high-density lipoprotein; Q, quartile; SBP, systolic blood pressure; SD, standard deviation.

**Table 2 geriatrics-11-00004-t002:** Associations of cardiometabolic index, body mass index and waist circumference with cardiometabolic multimorbidity.

Exposure	Events/ Total	Model 1		Model 2		Model 3	
		OR (95% CI)	*p*-Value	OR (95% CI)	*p*-Value	OR (95% CI)	*p*-Value
**Cardiometabolic index**							
Per 1 SD increase	197/3348	1.36 (1.21–1.51)	<0.001	1.26 (1.09–1.45)	0.002	1.25 (1.08–1.44)	0.003
Tertile 1 (0.06–0.37)	37/1116	ref		ref		ref	
Tertile 2 (0.38–0.72)	63/1116	1.75 (1.15–2.66)	0.009	1.44 (0.90–2.29)	0.13	1.40 (0.88–2.24)	0.16
Tertile 3 (0.73–6.40)	97/1116	2.75 (1.85–4.09)	<0.001	1.98 (1.15–3.41)	0.013	1.88 (1.09–3.25)	0.023
**Body mass index (kg/m^2^)**							
Per 1 SD increase	197/3348	1.47 (1.30–1.66)	<0.001	1.31 (1.14–1.49)	<0.001	1.28 (1.12–1.47)	<0.001
Tertile 1 (15.1–25.2)	39/1125	ref		ref		ref	
Tertile 2 (25.3–28.8)	60/1135	1.51 (0.99–2.28)	0.053	1.31 (0.85–2.01)	0.22	1.30 (0.85–2.00)	0.23
Tertile 3 (≥28.9)	98/1088	2.69 (1.84–3.94)	<0.001	1.95 (1.29–2.95)	0.001	1.88 (1.24–2.85)	0.003
**Waist circumference (cm)**							
Per 1 SD increase	197/3348	1.69 (1.47–1.94)	<0.001	1.49 (1.28–1.73)	<0.001	1.46 (1.25–1.71)	<0.001
Tertile 1 (60.5–88.8)	33/1122	ref		ref		ref	
Tertile 2 (88.9–100.3)	69/1119	2.31 (1.49–3.59)	<0.001	1.95 (1.24–3.04)	0.004	1.91 (1.22–2.99)	0.005
Tertile 3 (≥100.4)	95/1107	3.36 (2.17–5.18)	<0.001	2.27 (1.43–3.60)	0.001	2.16 (1.35–3.44)	0.001

CI, confidence interval; OR, odds ratio; SD, standard deviation. Model 1: Adjusted for age and sex. Model 2: Model 1 plus smoking status, alcohol consumption, systolic blood pressure, total cholesterol, high-density lipoprotein cholesterol, and handgrip strength. Model 3: Model 2 plus physical activity.

**Table 3 geriatrics-11-00004-t003:** Measures of risk discrimination upon addition of CMI, BMI, and WC to a CMM risk model containing established risk factors.

Measure of Discrimination	CMI	BMI	WC
C-index (95% CI): established risk factors	0.6892 (0.6500, 0.7285)	0.6892 (0.6500, 0.7285)	0.6892 (0.6500, 0.7285)
C-index (95% CI): established risk factors plus exposure	0.6924 (0.6528, 0.7319)	0.6941 (0.6551, 0.7331)	0.6992 (0.6603, 0.7382)
C-index change (*p*-value)	0.0032 (0.55)	0.0049 (0.46)	0.0100 (0.24)
*p*-value for difference in −2 log likelihood	0.004	<0.001	<0.001

BMI, body mass index; CI, confidence interval; CMI, cardiometabolic index; CMM, cardiometabolic multimorbidity; WC, waist circumference; established risk factors include age, sex, smoking status, alcohol consumption, systolic blood pressure, total cholesterol, high-density lipoprotein cholesterol, handgrip strength, and physical activity level.

## Data Availability

Data from the ELSA are available to the public to download from the UK Data Service at https://ukdataservice.ac.uk/ (accessed on 14 December 2024).

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
