# Peer review of "Cardiometabolic Index, BMI, Waist Circumference, and Cardiometabolic Multimorbidity Risk in Older Adults"

_geriatrics, 2025, doi:10.3390/geriatrics11010004_

Round 1

Reviewer 1 Report

Comments and Suggestions for Authors

The paper by Drs. Setor K. Kunutsor and Jari A. Lukkanen covers a topic that would be of great interest to readers of Geriatrics (ISSN 2308-3417). This work builds upon a growing body of literature in the area and provides a practical and relatively easily obtainable metric to provide further insight into the risk for cardiometabolic disease (and cardiometabolic multimorbidity (CMM)) in a large population cohort. The conclusions are consistent with the current literature and the findings from the current study. This work would represent a significant contribution to the available literature and would likely be well read and cited. The authors appropriately set the stage for the importance of assessing CMI and the development/prediction of CMM.

I have some minor revisions for the authors' consideration. 

It is recommended that the authors shorten the title. Currently, it is quite long and can be more succinctly and clearly described. "Prospective Associations of the Cardiometabolic Index With Cardiometabolic Multimorbidity and Its Predictive Utility Compared With BMI and Waist Circumference in the English Longitudinal Study of Ageing". 

The authors should include a hypothesis at the end of the introduction to reflect the hypothesis-driven nature of the work. The authors may want to enhance the introduction to better outline why they thought that the current determination of CMI may outperform the simple measurement of waist circumference (particularly in the context of age-related lipid changes). 

The study would benefit from a flow diagram to outline the number of dropouts and missing data. The authors are cautioned to temper statements related to the representativeness of this data and generalizability to other cohorts. 

The usage of acronyms (excessive) detracted somewhat from the readability of the work. 

The handgrip data is very interesting and likely warrants further attention (outside of being a covariate). Discussion of the handgrip values in the context of the threshold for functional dependance may be warranted. Its relationship to sarcopenia in older adults also warrants brief discussion. 

Also, further information is required on how participants were classified via physical activity. The usage of the term "sedentary" may be misleading. Are the authors referring to physically inactive or were there clients that were truly sedentary (such as those confined to bed rest). 

Please report the incidence (if any) of exercise-related adverse events. 

Further discussion of other potential co-variants not included is warranted. 

There is a lack of clarity related to cardiovascular disease in sections of the manuscript. A clear operational definition of CVD is likely warranted. 

Reviewer 2 Report

Comments and Suggestions for Authors

The revised manuscript, entitled “Prospective Associations of the Cardiometabolic Index With 2 Cardiometabolic Multimorbidity and Its Predictive Utility  Compared With BMI and Waist Circumference in the English 4 Longitudinal Study of Ageing” is generally interesting but suffers from some flaws.

The main limitation of this study is the lack of information on lipid-lowering therapy. Such treatment may substantially affect the predictive value of CMI.

Minor comments:

  1. In the tables, the age should be given as a mean and SD and minimum to maximum.
  2. All P values must be given to three decimal places.
  3. The IQR is the difference between Q3 and Q1, so either calculate the IQR or correct that the Q1 - Q3 is given in the description of the statistical analysis (TG values?).
  4. Please translate all abbreviations, e.g. PA in table 1
